# Examining the Indirect Effect of Online Gaming on Depression via Sleep Inequality and Anxiety—A Serial and Parallel Mediation Analysis

**DOI:** 10.3390/jcm11247293

**Published:** 2022-12-08

**Authors:** Tahani Alshammari, Sarah Alseraye, Aleksandra Rogowska, Nouf Alrasheed, Musaad Alshammari

**Affiliations:** 1Department of Pharmacology and Toxicology, College of Pharmacy, King Saud University, Riyadh 11451, Saudi Arabia; 2Clinical Pharmacy Department, King Fahad Medical City, Ministry of Health, Riyadh 12231, Saudi Arabia; 3Institute of Psychology, University of Opole, 45-040 Opole, Poland

**Keywords:** depression, anxiety, gaming disorder, sleep quality, mental health, first-year college students

## Abstract

Stress-related disorders are highly prevalent among first-year college students. Gaming disorder (GD) is an emerging disorder linked to physical and psychological consequences. We aimed to investigate the mechanism linking GD with anxiety, depression, and sleep disorders among first-year undergraduate students. Four hundred fifty-seven participants were recruited, and the survey included the Internet Gaming Disorder Scale Short-Form (IGDS9-SF), Generalized Anxiety Disorder-7 (GAD-7), Patient Health Questionnaire-9 (PHQ-9), and Pittsburgh Sleep Quality Index (PSQI). Our results showed that female students scored significantly higher than males in anxiety and depression. Furthermore, we found that depression is positively and strongly correlated to anxiety, and both are moderately associated with sleep quality. Gaming is positively related to depression, anxiety, and sleep quality. Interestingly, the health sciences tracks showed lower sleep quality than undergraduates from other tracks. There was a 64% variance in depression explained by many predictors, including anxiety, sleep quality, gaming, painkiller use, and gender. In addition, the mediation models showed that the association between gaming and depression is mediated indirectly by sleep quality, and sleep quality may be mediated directly by anxiety. The first year in college occurs at a critical developmental and professional stage, and our results highlight the need to establish support programs and conduct mental health educational workshops.

## 1. Introduction

Depression and anxiety are significant health burdens, and the rate of death is higher among depressed and anxious patients [1]. Economically, the severity of depression is linked to increased unemployment and reduced work performance [2]. At the global level, mental disorders such as anxiety and depression are common in undergraduate students, particularly first-year students [3,4,5]. We previously reported that undergraduate health science students exhibited a higher risk of developing anxiety and depression in the first wave of the pandemic [5]. We have also reported that the prevalence of depression is about 50% in cross-sectional settings during the second wave of COVID-19 [6]. Comparable findings were reported in a study of first-year Irish undergraduate students. Further, their results showed that about 30% of participants with a risk of depression experience suicidal ideation [7], indicating the immense burden of depression, especially in the young population.

The first year at university is considered a massive transition from late adolescence to adulthood, as students are expected to take on greater responsibilities [8,9]. According to the American Psychological Association (APA), around 35% of first-year university students are diagnosed with mental health disorders worldwide [9]. In comparison to other study years, first- and last-year students reported a higher prevalence of depression and anxiety [10]. These mental disorders have many negative consequences on students’ lives. Studies showed that depression and anxiety are associated with quality of life impairments [11,12,13], lower academic performance and GPA among undergraduate students [14,15], sleeping disorders [16], burnout, suicidal ideation [17], and higher dropout rates [18,19]. In addition, undergraduate students have a high prevalence rate of insomnia, which also affects their physical and mental health, academic achievement, and quality of life [16,20,21].

Over the last decade, many people of different ages consumed their leisure time in online gaming and are motivated to play these games for many reasons, including their feelings of challenge, socialization, and relaxation [22]. Gaming disorder (GD) is characterized by a predominant occupation with gaming behaviors over at least one year. GD was included in the 11th edition of the International Classification of Diseases (ICD-11) as a clinical condition, mainly when individuals’ social, educational, or occupational life aspects are impaired [23]. The global prevalence of GD was approximately 3% in meta-analysis studies [24,25] and reached up to 17% in a recent study conducted among Chinese adolescent game players [26]. In Saudi Arabia, a couple of studies have been conducted to examine the prevalence rate of GD, which was estimated at 5% among Saudi adolescents [27] and 8.8% among medical students at King Saud University [28].

Younger age was positively associated with a higher frequency of GD [29,30]. Online gaming and GD were also reported among college students [31,32,33]. It was estimated that almost half of college students play online or video games and about 4% of them have GD [33]. The risk of GD may be higher as they utilize the internet and digital devices during their study at university. In addition to that, due to the COVID-19 pandemic, many colleges have shifted their learning environment to a virtual learning environment in which the student utilization of the internet and technology may influence their online gaming consumption [34]. GD has many negative implications, particularly on the player’s mental health. Previous studies have shown that GD is associated with many mental health-related issues, such as stress [27,35], depression [30,35,36], anxiety [30,35], loneliness [37], suicidal ideation [38], and lower life satisfaction [35] as well as lower academic achievement [36]. Furthermore, GD may affect an individual’s physical health, as playing for a long time was associated with musculoskeletal symptoms [39] and headache [40] and led to the emergence of digital eye strain, a syndrome associated with the prolonged use of digital screens [41].

A scoping review showed that various factors of GD (including social isolation) interplay across the lifespan, leading to the deterioration of quality of life [42]. We aimed to examine the mechanisms of associations between anxiety, sleep quality, and online gaming to explain depression during the COVID-19 pandemic in the present study. Numerous studies indicate a high prevalence of depression, anxiety, and insomnia during the pandemic, in particular, among university students [43,44,45]. Moreover, insomnia, anxiety, and depression were intercorrelated among university students during COVID-19 [46]. Therefore, this is essential to understand the specific pattern of these associations in relation to online gaming, which also increased during the global time of crisis [47,48,49]. These associations were analyzed using mediation analysis. Mediation analyses were conducted previously in the same context as our research. A report analyzed the role of bedtime procrastination in mediating smartphone addiction, depression, and anxiety in Chinese participants, mostly undergraduates. Their findings indicated that participants with smartphone addiction are prone to procrastinating their bedtime, depression, and anxiety [50]. In another study, it was reported that anxiety and depression partially mediate insomnia and emotional stability [51]. Insomnia is linked to depression biologically [52,53] and psychologically [54,55]. In clinical settings composed of healthy adults, a three-night consecutive awakening was linked to a significant reduction in mood, suggesting a potential mechanism of the depression–insomnia relationship since a good mood promotes resilience to stress and strengthens coping mechanisms [56]. Additionally, anxiety and depression are highly comorbid [57]. About 45% of depressed individuals experience anxiety [58]. Therefore, it was reasonable to consider insomnia and anxiety as mediators for depression.

According to the biopsychosocial model of health, well-being is determined by various factors derived from the biological, psychological, and social areas of human life [59,60]. In the present study, we will consider the effect of the interplay between biological (gender), behavioral (game addiction, sleep quality), and psychological factors (anxiety) on depression as one of the indicators of subjective well-being.

Since online gaming, internet and smartphone addiction are related to each other [61,62,63], sharing criteria of behavioral addiction, we can assume that similar mediation to previous studies will be found in our research. This study aims to examine the mediating role of sleep quality and anxiety on the relationship between gaming addiction and depression among first-year students from King Saud University. Since female sex is a significant risk factor for anxiety, depression, and insomnia [64,65,66], while gaming disorder is more prevalent among men [67,68,69], gender differences will be also considered in this study.

## 2. Materials and Methods

### 2.1. Participants

A total of 457 male and female students participated in the study. The included participants were students who were studying at King Saud University for their first year of different colleges which included the health track, nursing track, science track, humanities track, business administration track, and applied and community services track.

### 2.2. Measurements

#### 2.2.1. Study Design and Procedure

A cross-sectional study was conducted among first-year female and male students from King Saud University from the period of 23 January to 27 February 2022. An online survey was created via Google Forms in two languages, Arabic and English, after some amendments after the pilot of 7 students and distributed among the participants by using convenient sampling technique through Twitter, WhatsApp, Telegram, etc. A consent message indicating that the participant agreed to participate in the study and the response is confidential and completely voluntary along with a clear indication of the project aims and descriptions was stated at the beginning of the survey. Ethical approval was granted by the Institutional Review Board at King Saud University in Riyadh, Saudi Arabia (Ref. No. 22/0055/IRB).

#### 2.2.2. Measurements

##### Depression

The PHQ-9 is a nine-item questionnaire used to measure depression symptoms following the criteria of Diagnostic and Statistical Manual of Mental Disorders, Fourth Edition (DSM-IV). It includes 9 items that measure the frequency of depression over the last two weeks rated on a scale of 0 to 3 (0 = Not at all, 1 = Several days, 2 = More than half the days, 3 = Nearly every day). Total scores were summed, and depression severity was interpreted as normal (0–4), mild (5–9), moderate (10–14), moderately severe (15–19), and severe (20–27) [70]. Both English [71] and Arabic [72] versions of this tool were used in the study.

##### Anxiety

The GAD-7 scale is a validated seven-item scale used widely and easily to screen for anxiety in general and research settings [73]. The GAD-7 scale uses a four-point Likert scale from 0 (not at all) to 3 (every day) with total scores ranging from 0 to 21; higher scores represent higher grades of anxiety. The total scores of 5, 10, and 15 are taken as the cut-off points for mild, moderate, and severe anxiety, respectively. Both English [74] and Arabic [75] versions of this tool were used in the study.

##### Sleep Quality

The PSQI is a valid scale used for the assessment of sleep habits during the last month. The questionnaire included 19 items divided into seven components (rated from 0 to 3) to assess different variables related to sleep quality [76]. The seven components are C1 (Subjective sleep quality), C2 (Sleep latency), C3 (Sleep duration), C4 (Sleep efficiency), C5 (Sleep disturbance), C6 (Use of sleep medication), and C7 (daytime dysfunction). The total PSQI score ranges from 0 to 21, with higher scores representing worse sleep quality. Both English [77] and Arabic [78] versions of this tool were used in the study.

##### Gaming

The IGDS9-SF is one of the psychometric tools used commonly for the assessment of internet GD severity [79,80]. The IGDS9-SF consists of 9 items developed to assess the severity of the internet GD over 12 months, a criterion suggested by the APA in the DSM-5 for the internet GD diagnosis [81,82,83]. IGDS9-SF has many advantages including great utility in research and clinical settings, as it requires less time [81] and robust psychometric properties (internal consistency and validity) [83]. The IGDS9-SF scale uses a five-point Likert-type scale: Never (1), Rarely (2), Sometimes (3), Often (4), and Very often (5). The total score will range from 9–45, where a higher score indicates a higher level of GD. Both English [80] and Arabic [84] versions of this tool were used in the study.

##### Demographics

The demographic characteristics assessed in this study consist of participants’ age, gender, marital status, number of family members at home, current common track, the first generation to receive higher education, pain killer use, and eye lubricant use.

#### 2.2.3. Statistical Analysis

Descriptive statistics were conducted, including frequencies (*n*), percentage (%), mean (*M*), median (*Mdn*), standard deviation (*SD*), skewness, and kurtosis. Reliability of PHQ-9, GAD-7, PSQI, and IGDS9-SF was assessed in the total sample (*n* = 457) using Cronbach’s α coefficient of internal consistency. Gender (women, men) and study track (health sciences and nursing tracks, other tracks) differences were examined using the independent samples Student’s *t*-test while the association between depression, anxiety, sleep quality, and gaming was conducted using Pearson’s correlation. Hierarchical multiple linear regression analysis was conducted for depression as an explained dependent variable, and demographics (gender, number of family members at home, the transition of education in the family, and faculty), painkillers use, eye lubricant use, gaming, sleep quality, and anxiety were used as independent predictor variables. Finally, two models of mediation were examined, parallel Model 1 and serial Model 2, to find a mechanism explaining the association between gaming and depression among undergraduate students in Saudi Arabia during the COVID-19 pandemic. The mediating effect of sleep quality and anxiety on the relationship between gaming and depression was examined with the maximum likelihood estimation method and bias-corrected percentile bootstrap technic to assess a 95% confidence interval. Statistical analyses were conducted using JASP 0.16.1 software for Windows except the alluvial diagrams, which were performed using JAMOVI software for Windows. The serial model of mediation was performed using PROCESS macro v.4.0 for IBM SPSS software for Windows (Model 6 was applied).

## 3. Results

### 3.1. Participant Characteristic

The total sample consisted of 457 first-year students aged between 17 and 29 (*M*_Age_ = 18.78, *SD*_Age_ = 1.27). The demographic characteristics of the sample are presented in Table 1. Among participants, women (77%), students of the science track (34%), and individuals with parents with higher education (81%). Given the status of relationship, only one student was married while the rest of the sample declared themselves single. Most students did not use painkillers (59%) or eye lubricants in drops (70%).

### 3.2. Prevalence of Depression, Anxiety, Sleep Difficulties, and Gaming Addiction

The prevalence of the risk of anxiety at moderate or severe levels presented as 54% of undergraduates (*n* = 249) while the risk of depression presented at the same levels 58.43% (*n* = 267) of the total sample (*n* = 457). A total of 87 students (19.04%) showed no symptoms of anxiety, 121 (26.48%) showed mild symptoms, 123 (26.92%) showed moderate symptoms, and 126 (27.57%) showed severe symptoms. Regarding depression, no symptoms were found in 17.72% of the sample (*n* = 81), mild symptoms were found in 23.85% (*n* = 109), moderate symptoms were found in 25.60% (*n* = 117), and severe symptoms were found in 32.82% of undergraduates (*n* = 150). Poor sleep quality (PSQI > 7) presented in 62.58% of the student population (*n* = 286) while gaming addiction risk was shown in 37.64% of students (*n* = 172).

### 3.3. Descriptive Statistics

Initially, the descriptive statistics were assessed using a range of scores: mean (M), standard deviation (SD), median (Mdn), skewness, kurtosis, and Cronbach’s α (Table 2). Total scores of anxiety (GAD-7), depression (PHQ-9), gaming (IGDS9-SF), and sleep quality (PSQI) showed appropriate characteristics for parametric tests (skewness and kurtosis ranged between +1 and −1). Therefore, Student’s *t*-test, Pearson’s correlation, and linear regression analysis were conducted in the following steps.

### 3.4. Group Differences

Gender differences in anxiety, depression, gaming addiction, and sleep quality were examined using the independent samples Student’s *t*-test (Table 3). Women scored significantly higher than men in anxiety and depression (with a small effect size) while no gender effect was found on sleep quality and gaming.

The Student’s *t*-test was applied to assess their study major—whether it was in the applied and community service, business administration, health, humanities, nursing, or science tracks (Table 4).

Students who selected health sciences or nursing as a track showed significantly lower sleep quality scores (but the effect size was small) than undergraduates who chose other study tracks to study, such as applied and community service, business administration, humanities, or science. No track differences were found for anxiety, depression, and gaming.

### 3.5. Associations between Variables

Pearson’s correlations were performed to examine correlations between depression, anxiety, sleep quality, and gaming (Figure 1). Depression is positively and strongly correlated to anxiety. Anxiety and depression are positively and moderately associated with sleep quality. Gaming is positively but weakly related to depression, anxiety, and sleep quality.

Hierarchical multiple linear regression analysis was conducted for depression (Table 5). Demographic variables, such as gender, number of family members at home, the transition of education in the family, and faculty, were included in Model 1. These variables explained only 3% in depression, and female gender was a significant predictor. Both painkiller and eye lubricant use were added to Model 2 of the regression, explaining about 9% more variance in depression. However, only gender and painkiller use were significant predictors of depression. When gaming was included in Model 3, gender, painkiller use, and gaming were significantly related to depression. The explained variance increased by about 4% in Model 3. Model 4 contained additional sleep quality, which was a significant predictor of depression, explaining 24% of its variance. In the last step of regression analysis, anxiety was included in Model 5 beside all previously mentioned variables. Anxiety was strongly associated with depression, which explains the next 24% of its variance. The other significant predictors were sleep quality, gaming, painkiller use, and gender. The total Model 5 (with all variables) explains 64% of depression variance.

### 3.6. Mediation Analysis

The mediational Model 1 (parallel mediation) and Model 2 (serial mediation) were performed for depression as an outcome, gaming as a predictor variable, and both sleep quality and anxiety as mediators. In addition, gender and painkiller use were included in the model of mediation as confounders. The results of the mediation analysis for Model 1 are presented in Figure 2 and Table 6. Sleep quality and anxiety partially mediate the relationships between gaming and depression. Sleep quality can be predicted by gaming directly (total effect) and indirectly through sleep quality and anxiety. The model explains 64% of depression variance.

Model 2 of mediation showed that depression can be predicted by gaming indirectly through sleep quality in a simple partial mediation (Table 7 and Figure 3). The other possible mechanism leads partially from gaming via sleep quality and anxiety to depression, considering a serial mediation model. However, the path between gaming and anxiety was insignificant, so anxiety cannot mediate the gaming–depression association. In contrast, sleep quality fully mediates the indirect effect of gaming addiction on anxiety.

## 4. Discussion

The present study examined the mechanisms of associations between depression, anxiety, sleep quality, and online gaming. We found that female students scored significantly higher than males in anxiety and depression. Furthermore, this study found that depression is positively and strongly correlated to anxiety. Both anxiety and depression are positively and moderately associated with sleep quality, and gaming is positively but weakly related to depression, anxiety, and sleep quality. Furthermore, the health sciences and nursing tracks showed significantly lower sleep quality than undergraduates from other tracks. Our multiple regression analysis indicated that about a 64% variance of depression is explained by many predictors, including anxiety, sleep quality, gaming, painkiller use, and gender. In addition, the mediation models showed that the association between gaming and depression is mediated indirectly by sleep quality, and sleep quality may be mediated directly by anxiety.

Our descriptive analysis revealed that the study participants exhibited a high risk of developing mental disorders. Indicators of anxiety, depression, and GD were present in around one-third of the participants. Risks of insomnia were reported to be in the majority of the sample. Insomnia is defined as a persistent condition of reduced sleep quantity or quality due to difficulties initiating or maintaining sleep [85]. At the same time, anxiety is an affective condition characterized by nervousness, worry, and tension. It is usually accompanied by physical symptoms, including but not limited to dizziness, sweating, and elevated blood pressure [86]. Multiple notions could interpret the mean scores from our descriptive statistics findings. First, this could be attributed to the fact that the first year of university is a determinate transition state [9] as well as stressors related to class re-entry following the outbreak of the COVID-19 pandemic. Additionally, some changes might have been adapted since the start of the pandemic [87]. For instance, people have been spending more time on screens [88], and these changes might have adapted even after the COVID-19 pandemic. Other factors that might contribute to these findings are recent changes in the young population’s lifestyle. Notably, multiple observations were reported on excessive gamer lifestyles, including increased nighttime activity, altered sleep–wake cycles, and altered eating habits [89].

Our data suggest that first-year female students have a higher risk of developing depression. Gender difference in depression is considered one of the most robust phenomena in psychiatry and psychology, with more depressed women than men [90]. In line with this, a systematic review and meta-analysis conducted among healthcare workers during the COVID-19 pandemic found that rates of anxiety and depression were higher for female healthcare workers and nursing staff compared to their male colleagues [91]. Another report found that more than half of healthcare undergraduate female students and a third of male students have at least moderately high test anxiety [92]. Overall, sexual dimorphism is well-acknowledged in mood disorders [93].

Our study track analyses indicated that first-year students in health sciences and nursing tracks showed a trend toward lowered sleep quality and a significant elevation in developing anxiety and depression. First-year university students face numerous stressors, including academic requirements, social adjustments, and time management. Medical students, in particular, may face further challenges, for example, the large study workload, commitment to assessments, and the pressures along with the liability of a clinical environment [94]. Being a member of the healthcare profession increases the risk of mood disorders. A systematic review and meta-analysis done among healthcare workers during the COVID-19 pandemic showed that at least one in five healthcare professionals report symptoms of depression and anxiety [95]. In addition, a recent meta-analysis revealed that depression affects approximately one-third of medical students worldwide [96]. Overall, evidence has indicated that the risk of developing anxiety in medical students is significant compared with nonmedical students [97].

Our regression analyses indicate that anxiety strongly correlates with depression. Our findings verify existing knowledge. For instance, in a sample of Australian university students, the anxiety–depression comorbidity was more than a third, and in fact, about four times that for anxiety or depression alone. This comorbidity highlights the profound clinical consequences of anxiety and depression [98]. In line with this, a multicenter-based report indicated a bidirectional association between (1) anxiety and depression promoting insomnia; and (2) insomnia elevating the risk of anxiety and depression [99]. Additionally, a prospective longitudinal study reported that social anxiety disorder during adolescence is a crucial predictor of subsequent depressive disorders. The study findings indicated that social anxiety–depression comorbidity during adolescence is linked to a subsequent severe major depressive disorder [100].

Our data show that depression is linked to using painkillers and eye lubricants. Following the inclusion of GD in the WHO classification of diseases (ICD-11), studies have examined some of their consequences on general health. These issues may include musculoskeletal conditions, eyes problems, exhaustion, and headaches [39,101], which may increase the utilization of over-the-counter medications to relieve these symptoms, such as painkillers and lubricating eye drops. In line with this, a cross-sectional study in medical school students showed that GD is associated with headaches and less proficiency in time management. A higher prevalence of GD is linked to skipping classes and cognition-related issues [102]. Another study showed that GD is associated with performance-enhancing drugs; the report indicated that two out of five gamers consumed stimulants, including prescribed and over-the-counter products [103]. Moreover, a study has shown that playing a video game is associated with anatomical changes in cortical thickness, a brain region critical for decision making and cognitive functions [104].

In both mediation models, the parallel and serial, GD was the predictor variable, sleep quality and anxiety were mediators, and depression was the outcome. At the same time, gender, and painkiller use were included as confounders. Our findings from the parallel mediation indicate that GD and depression could be partially mediated by poor sleep quality and anxiety. These results provide a mechanistic link between anxiety, sleep quality, and gaming behavior contributing to depression. Additionally, the parallel mediation analysis suggests that sleep quality can be predicted by gaming directly (total effect) and also indirectly through sleep quality and anxiety. In support of this, a systematic review showed that problematic internet use is associated with poor sleep quality [105].

Although technology is evading, most studies conducted on undergraduate students examining psychological stressors discuss the impact of internet addiction [63,106,107,108,109] rather than online gaming, even though a significant portion of the young population using technology are directed toward online gaming [110]. Predictors of GD were suggested in many studies, including the gender and the age group of the player. Many studies agreed that male gender was positively associated with more GD compared to female gender [26,28,29,30,111]. In addition, the purpose of playing online games was different among the two genders, as playing for the purpose of passing time was higher with females whereas more males play for the feeling of achievement and making friends [111].

A study in the Pakistani general population reported that GD was significantly linked to poor sleep quality [112]. Similar findings were observed in Indonesian students [113]. In addition, a study that examined sleep duration in Japanese male adolescent students reported that more than eight hours of sleep duration per night is associated with lower depression/anxiety risks [114]. Interestingly, a systematic review discussed the biological consequences of gaming addiction. The study examined neuroimaging evidence of neurocircuitry changes that occurred with gaming addiction and compared it with neurological changes with drug abuse [115], indicating that GD has biological hallmarks of traditional substance-related addictions. In line with this, a previous report examined reward circuitry using functional magnetic resonance imaging during video gaming. Participants were mainly young adults with around eight hours per week of experience in video gaming. The study findings indicated that the orbitomedial prefrontal cortex and the anterior putamen were activated differentially during winning and losin, indicating complex acquisition of reward circuitry while video gaming [116]. Additionally, the level of dopaminergic release is affected by video gaming [117].

A recent review reported that GD is linked to structural and functional changes in neuronal populations, particularly the fronto-striatal region. This region is physiologically implicated in modulating both compulsive behavior and attention. Further, the core pathology of GD includes psychological dependence. Additionally, individuals with GD exhibited personal features similar to traditional substance-related addicts, including the reduced personal capacity to evaluate risks, aggression, and impulsivity [89]. In line with this, another study underlined the addictive potential of problematic online gaming compared to other online activities. Further, it suggests that comorbidities of psychopathologies or addictions are relatively common. In fact, it is viewed as a typical rather exceptional condition [118], indicating the link between GD, insomnia, and depression is beyond passing the time, and it suggests the association is due to psychological addiction.

On the other hand, the serial model showed that depression could be predicted by gaming indirectly by sleep quality in a simple mediation. This could be attributed to gaming to sleep quality and anxiety then to depression. However, a direct path between gaming and anxiety was insignificant, suggesting anxiety cannot mediate the gaming–depression association. A study showed that excessive use of social networks is associated with poor sleep quality and elevated everyday cognitive failures. The study findings indicated that the use of social networks–cognitive failures is mediated by sleep quality [115].

In line with this, studies showed that online gaming is related to sleep deprivation and decreased sleep quality [119,120,121]. Furthermore, gaming addiction increases symptoms of anxiety and depression [61,62,69,122]. Insomnia increases the risk of both depression and anxiety [123,124,125]. Moreover, a systematic review suggests insomnia is bidirectionally related to anxiety and depression [126,127]. Insomnia is linked to depression biologically [52,53] and psychologically [54,55]. In clinical settings composed of healthy adults, a three-night consecutive awakening was linked to a significant reduction in mood. Bai et al. [124] evidenced that insomnia plays an essential role in the well-being of people during the COVID-19 pandemic, and prevention of insomnia is critical to reducing anxiety and depression. Insomnia and anxiety were found as predictors of depression previously [128].

On the other hand, a good mood promotes resilience to stress and strengthens coping mechanisms [56], suggesting a potential mechanism of the depression–insomnia relationship. Additionally, anxiety and depression are highly comorbid [57]. About 45% of depressed individuals experience anxiety [58]. However, research indicated that anxiety precedes depression and contributes to its severity [129]. Therefore, it was reasonable to consider insomnia and anxiety as mediators for the relationship between gaming addiction and depression.

A previous study indicated that delaying bedtime can indirectly affect anxiety and depression via poor sleep quality [130]. Some potential reasons for procrastination at bedtime are online gaming and excessive mobile phone or internet use. Indeed, several studies found a partial mediating effect of sleep quality on the relationship between problematic online behavior (i.e., internet addiction, social media, or mobile phone overuse) and well-being (especially depression) among university students from Algeria [131], China [132,133], and Nepal [134]. Moreover, Yu et al. [135] found a mediating role between insomnia and depression in the association between internet gaming disorder and suicidal ideation among Chinese adolescents. Furthermore, a recent study found both a serial and parallel mediation of anxiety, stress, and sleep quality on the relationship between internet addiction and depression among Serbian medical students [136,137].

As far as we know, this is the first report to explore the mechanistic effect of GD on insomnia, anxiety, and depression in first-year students. Moreover, given that exposure to the COVID-19 pandemic has altered mental health, this study is an updated measurement following the return to the typical lifestyle we used to have. Additionally, few studies have examined the impact of GD on the Saudi community. This report is updated and the first to analyze the implications of GD in first-year undergraduate students.

Our findings highlight the need for some interventions. For instance, first-year students, particularly health and medical students, need prioritization to increase awareness of the impact of good sleep quality and GD. Second, given that GD is a key mediator of poor sleep quality and depression, some actions should be taken to help students reduce time spent on gaming and the internet and improve their sleep quality, for instance, conducting a periodical educational workshop about these stressors. In addition, our findings highlight the need to implement a regulation restricting online gaming. A previous experience was implemented in South Korea [138]. Similarly, in China, a midnight patrol was initiated recently where players have a limited number of games after midnight [139].

### Study Limitations

Although the study findings are significant, some limitations exist. First, the study design is cross-sectional; further verification can be added by conducting a longitudinal study. Second, most participants were females; thus, we may not generalize the study conclusions. Third, the study was conducted at a single center. Moreover, the study was a self-reported survey, and thus reporter bias might exist.

## 5. Conclusions

Mental health problems can affect students’ academic performance and overall health [14]. In addition, online gaming is prevalent in college students and can lead to many mental health problems and affect students’ lives, health, and performance if not appropriately addressed.

## Figures and Tables

**Figure 1 jcm-11-07293-f001:**
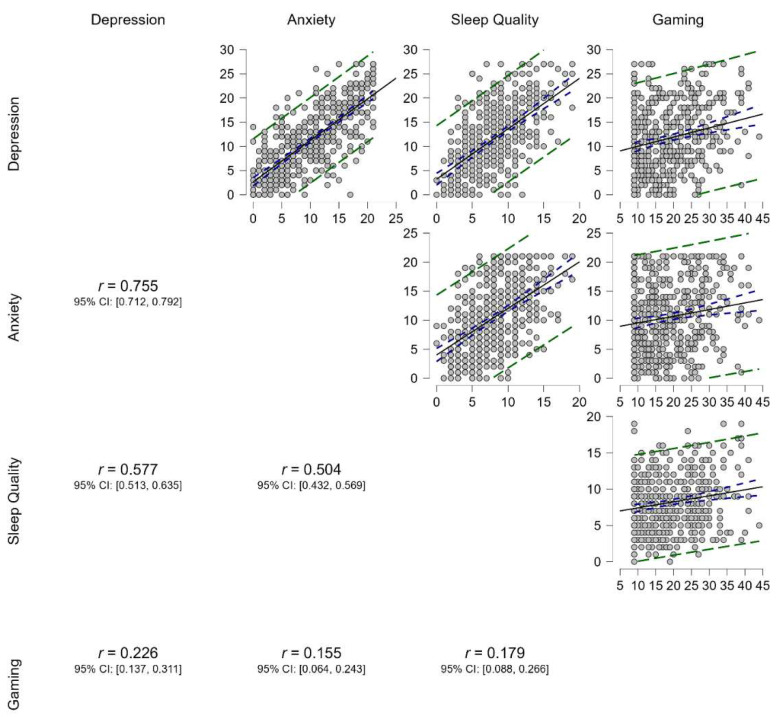
Pearson’s correlation between depression, anxiety, sleep quality, and gaming in a sample of undergraduates (*n* = 457). Note: All correlations are significant at *p* < 0.001. The solid gray line represents the regression slope; the purple dashed line shows a 95% confidence interval; and the green dashed line means 95% prediction intervals.

**Figure 2 jcm-11-07293-f002:**
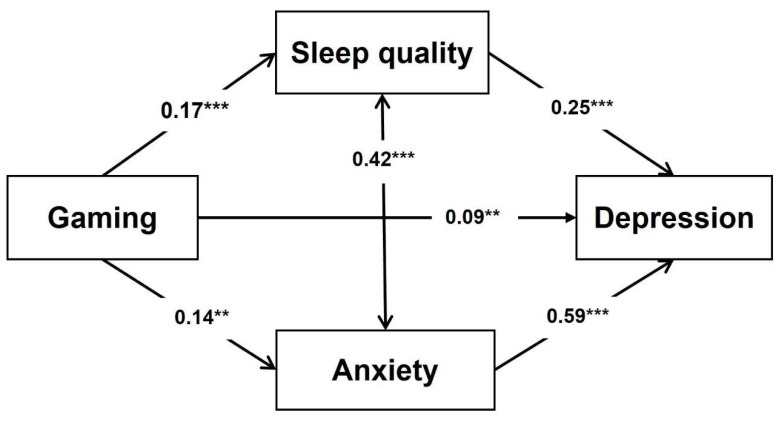
Mediating Model 1 for depression (parallel mediation). Note: The numbers on the paths are standardized β coefficients. ** *p* < 0.01, *** *p* < 0.001.

**Figure 3 jcm-11-07293-f003:**
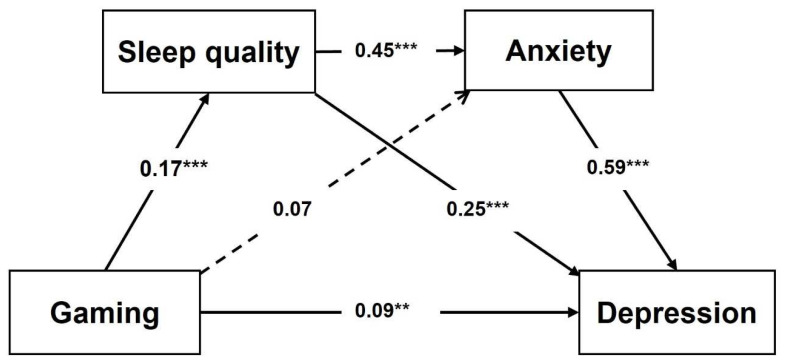
Mediating Model 2 for depression (serial mediation). Note: The numbers on the paths are standardized β coefficients. ** *p* < 0.01, *** *p* < 0.001.

**Table 1 jcm-11-07293-t001:** Demographic characteristics of the sample (*n* = 457).

Variable	*n*	%
Gender		
Female	352	77.02
Male	105	22.98
First year of common track		
Applied and community service	6	1.31
Business administration	125	27.35
Health	106	23.20
Humanities	35	7.66
Nursing	28	6.13
Science	157	34.35
First generation to receive higher education		
No	372	81.40
Yes	85	18.60
Painkiller use last week		
1 day	85	18.60
2 days	51	11.16
3 days	34	7.44
4 days	11	2.41
5 days	1	0.22
6 days	1	0.22
7 days	6	1.31
I did not use painkillers	268	58.64
Painkiller use per day		
1 time	103	22.54
2 times	72	15.76
3–4 times	14	3.06
5–6 times	2	0.44
Not applicable	266	58.21
Eye drop lubricant use last week		
1 day	39	8.53
2 days	26	5.69
3 days	19	4.16
4 days	13	2.85
5 days	6	1.31
6 days	5	1.09
7 days	28	6.13
I did not use eye drops lubricants	321	70.24
Eye drop lubricant use per day		
1 time	75	16.41
2 times	44	9.63
3–4 times	20	4.38
5–7 times	7	1.53
8–12 times	1	0.22
Not applicable	309	67.62

**Table 2 jcm-11-07293-t002:** Descriptive statistic (*n* = 457).

	Range							Cronbach’s α
Variable	Min.	Max.	M	SD	Mdn	Skewness	Kurtosis	
Anxiety (GAD-7)	0	21	10.55	6.03	10	0.11	−1.05	0.91
Depression (PHQ-9)	0	27	11.71	6.84	11	0.28	−0.73	0.88
Gaming (IGDS9-SF)	9	44	18.92	8.13	18	0.67	−0.32	0.87
Sleep quality (PSQI)	0	19	8.16	3.79	8	0.35	−0.33	0.69

**Table 3 jcm-11-07293-t003:** Student’s *t*-test for gender differences.

Variable	Women (*n* = 352)		Men (*n* = 105)		*t* (455)	*p*	Cohen’s *d*
Anxiety (GAD-7)	11.00	6.12	9.03	5.50	2.96	0.003	0.33
Depression (PHQ-9)	12.33	6.94	9.62	6.08	3.61	<0.001	0.40
Gaming (IGDS9-SF)	8.25	3.81	7.85	3.72	0.95	0.344	0.11
Sleep quality (PSQI)	18.61	8.37	19.94	7.24	−1.48	0.140	−0.16

**Table 4 jcm-11-07293-t004:** Student’s *t*-test for study track differences.

Variable	Other Faculties (*n* = 323)		Health & Nursing (*n* = 134)		*t* (455)	*p*	Cohen’s *d*
Anxiety (GAD-7)	10.59	6.09	10.43	5.91	0.272	0.785	0.028
Depression (PHQ-9)	11.82	6.87	11.43	6.79	0.566	0.572	0.058
Gaming (IGDS9-SF)	8.21	3.80	8.02	3.79	0.482	0.630	0.050
Sleep quality (PSQI)	19.60	8.15	17.25	7.89	2.833	0.005	0.291

**Table 5 jcm-11-07293-t005:** Hierarchical multiple linear regression for depression (*n* = 457).

Variable in the Model	Model 1	Model 2	Model 3	Model 4	Model 5
Gender	0.17 ***	0.11 *	0.12 **	0.12 **	0.07 *
Number of family members at home	0.06	0.05	0.04	0.05	0.02
First generation to receive higher education transition	−0.03	−0.02	0.01	0.00	0.02
Health sciences and nursing tracks	−0.02	−0.04	−0.01	0.00	0.00
Painkiller use		0.28 ***	0.26 ***	0.17 ***	0.07 *
Eye lubricant use		0.08	0.08	0.04	0.00
Gaming			0.21 ***	0.13 **	0.09 **
Sleep quality				0.51 ***	0.25 ***
Anxiety					0.58 ***
*R* ^2^	0.03	0.12	0.16	0.40	0.64
Δ*R*^2^	0.03	0.09	0.04	0.24	0.24
Δ*F*	3.80 **	22.00 ***	23.56 ***	181.27 ***	293.37 ***
*df* for Δ*F*	4.452	2.450	1.449	1.448	1.447

Note: Coding for categorical variables: gender (women = 1, men = 0), education transition (no = 1, Yes = 0), health sciences or nursing track (yes = 1, no = 0), painkiller using (yes = 1, no = 0), eye lubricant using (yes = 1, no = 0). * *p* < 0.05, ** *p* < 0.01, *** *p* < 0.001.

**Table 6 jcm-11-07293-t006:** Model 1 of mediating effects of sleep quality and anxiety on the relationship between gaming and depression in the sample of undergraduates (*n* = 457).

									95% Bca CI	
Effect	Variables in the Model					β	SE	z	LL	UL
D	Gaming	→	Depression			0.09	0.03	3.14 **	0.032	0.150
I	Gaming	→	Sleep quality	→	Depression	0.04	0.01	3.30 ***	0.016	0.070
I	Gaming	→	Anxiety	→	Depression	0.09	0.03	3.19 **	0.033	0.138
C	Sleep quality	↔	Anxiety			0.42	0.05	8.98 ***	0.347	0.511
T	Gaming	→	Depression			0.22	0.04	4.99 ***	0.135	0.298

Note: D = Direct effect, I = Indirect effect, C = Covariance, and T = total effect; ** *p* < 0.01, *** *p* < 0.001.

**Table 7 jcm-11-07293-t007:** Model 2 of mediating effects of sleep quality and anxiety on the relationship between gaming and depression in the sample of undergraduates (*n* = 457).

											95% Bca CI	
Effect	Variables in the Model							β	SE	t	LL	UL
D	Gaming	→	Depression					0.08	0.02	3.12 **	0.028	0.124
I	Gaming	→	Sleep	→	Depression			0.04	0.01	7.49 ***	0.016	0.071
I	Gaming	→	Anxiety	→	Depression			0.04	0.03	1.71	−0.009	0.088
I	Gaming	→	Sleep	→	Anxiety	→	Depression	0.44	0.01	17.37 ***	0.018	0.072
T	Gaming	→	Depression					0.18	0.04	4.97 ***	0.110	0.254

Note: D = Direct effect, I = Indirect effect, T = total effect; ** *p* < 0.01, *** *p* < 0.001.

## Data Availability

Data are available upon reasonable request.

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
