# Peer review of "Examining the Indirect Effect of Online Gaming on Depression via Sleep Inequality and Anxiety—A Serial and Parallel Mediation Analysis"

_jcm, 2022, doi:10.3390/jcm11247293_

Round 1

Reviewer 1 Report

I read the article with great interest, and judge it to be scientifically valuable and worthy of publication. The authors analyze the current problems of functioning of college students: involvement in online games and depression. The research problem and hypotheses are set correctly. The selection of measurement tools is adequate. Questionnaires used to measure the risk of depression, anxiety, sleep quality and severity of online gaming disorder (GD), were constructed according to APA criteria: DSM-IV and DSM5.
It is important that the authors included the gender of the subjects as a possible risk factor for the variables studied.

Comments_suggestions:
1. In the theoretical part, an important addition would be a concise description of the psychological indicators of the disorders described: gaming disorder (GD) and depression, anxiety and sleep disorders. The terms depression, anxiety and sleep disorders are increasingly common in everyday language, it would be worthwhile for the article to include information on the criteria proposed by the Diagnostic and Statistical Manual of Mental Disorders according to the APA - for the reader outside the discipline of psychology.
2. In the theoretical part: mediation models should have a better theoretical justification: why did the authors propose such an arrangement of relationships between variables? This is partly explained in the discussion of the results, but it would be good if the mediation research hypotheses were derived from theoretical assumptions.

3. It seems to me that the analyses of mean scores from descriptive statistics are worth a concise interpretation, because they show a large percentage of students with high risk indicators of disorders: anxiety 126 persons (27.57%), depression 150 persons (32.82%), sleep quality 286 persons (62.58%), risk of game addiction 172 persons (37.64%).

Author Response

Thank you for your constructive comment. Our responses for each point are attached.

Reviewer 2 Report

This highly original paper evaluates online games for the association between sleep and depression. The study is focused on Polish first-year university students, an age group that is socially and stressfully vulnerable. The study is cross-sectional rather than longitudinal, with a slightly higher proportion of girls, but this bias is also noted as a limitation at the end. Overall, the paper is evaluated as having appropriate content. Here are some questions and comments from the reviewers The reviewers expect the authors to respond to all items in a clear and concise manner.

Major1

The definition of the diagnosis of gaming disorder needs to be properly described in the introduction. The DSM-5 and ICD-11 are good references for the diagnosis of Internet gaming disorder.

Major 2

The relationship between sleep cycles and late night gaming is today discussed as a new lifestyle related disease. How about adding this point to the list? Please cite to as follow paper: J.Clin. Med 2022, 11, 4566. https://doi.org/10.3390/jcm11154566

Major 3

One of the reasons why gaming addiction is associated with sleep and depression is the idea that gaming addiction is not just a way to pass the time, but a psychological addiction. You may want to refer to the following paper. J. Behav. Addict. 2017, 6, 133-141   Front Psychol. 2018, 28,787.

Minor 1

The past decade has seen a description of efforts to address gaming disorders in China and Saudi Arabia. A country with even stricter initiatives is South Korea. In Korea, there is a system commonly known as the Cinderella Law, which stops online gaming at midnight. How about referring to the citation?

Minor2

There is little description of the medical mechanisms related to gaming disorder and depression. There are many papers on MRI studies of adolescents that show changes in prefrontal and reward system volumes. Why don't you look them up and cite them in your discussion.

Peer reviewers are waiting to hear back from the authors.

Best regards, 

Dr. Imataka

Author Response

(The authors gave the same response as above.)

Round 2

Reviewer 2 Report

The authors have responded appropriately to all the reviewers' comments. The authors have also revised the text according to the responses. The content of the paper is now clearer and easier for readers to understand. The authors should double-check that the final figures in the tables are correct before publication.

Best regards,

Dr,Reviewer

Author Response

thank you for your support, tables' numbering is revised 
